# Comparative Pharmacokinetics of Gentamicin C_1_, C_1a_ and C_2_ in Healthy and Infected Piglets

**DOI:** 10.3390/antibiotics13040372

**Published:** 2024-04-18

**Authors:** Eun-Young Kim, Tae-Won Kim, Elias Gebru Awji, Eon-Bee Lee, Seung-Chun Park

**Affiliations:** 1Laboratory of Veterinary Pharmacokinetics and Pharmacodynamics, Institute for Veterinary Biomedical Science, College of Veterinary Medicine, Kyungpook National University, Daegu 41566, Republic of Korea; key0@korea.kr (E.-Y.K.); eliawji1@gmail.com (E.G.A.); 2College of Veterinary Medicine and Institute of Veterinary Science, Chungnam National University, 99 Daehak-ro, Yuseong-gu, Daejeon 34134, Republic of Korea; taewonkim@cnu.ac.kr; 3Veterinary Drugs & Biologics Division, Animal and Plant Quarantine Agency, Ministry of Agriculture, Food and Rural Affairs, Gimcheon 39660, Republic of Korea; 4Cardiovascular Research Institute, Kyungpook National University, Daegu 41566, Republic of Korea

**Keywords:** gentamicin, pharmacokinetics, pharmacodynamics, piglets, *Actinobacillus pleuropneumoniae*, *Pasteurella multocida*

## Abstract

Gentamicin, an aminoglycoside antibiotic, is a mixture of therapeutically active C_1_, C_1a_, C_2_ and other minor components. Despite its decades-long use in pigs and other species, its intramuscular (IM) pharmacokinetics/pharmacodynamics (PKs/PDs) are unknown in piglets. Furthermore, the PKs of many drugs differ between healthy and sick animals. Therefore, we investigated the PKs of gentamicin after a single IM dose (10 mg/kg) in healthy piglets and piglets that were intranasally co-infected with *Actinobacillus pleuropneumoniae* and *Pasteurella multocida* (PM). The plasma concentrations were measured using validated liquid chromatography/mass spectrometry. The gentamicin exposure was 36% lower based on the area under the plasma concentration–time curve and 16% lower based on the maximum plasma concentration (C_max_) in the infected piglets compared to the healthy piglets, while it was eliminated faster (shorter half-life and larger clearance) in the infected piglets compared to the healthy piglets. The clearance and volume of distribution were the highest for the C_1_ component. C_1_, C_1a_ and C_2_ accounted for 22–25%, 33–37% and 40–42% of the total gentamicin exposure, respectively. The PK/PD target for the efficacy of aminoglycosides (C_max_/minimum inhibitory concentration (MIC) > 10) could be exceeded for PM, with a greater magnitude in the healthy piglets. We suggest integrating this PK information with antibiotic susceptibility data for other bacteria to make informed antibiotic and dosage regimen selections against piglet infections.

## 1. Introduction

Gentamicin, a bactericidal aminoglycoside antibiotic, is a mixture of pharmacologically active components, mainly C_1_, C_1a_ and C_2_ and other minor components or impurities [1,2]. The major components differ mainly in their methylation patterns on the 2-amino-hexose ring (Figure 1). Gentamicin C_1a_ is devoid of methyl groups on this ring, while both C_1_ and C_2_ have a methyl group at the 6′ position. C_1_ further distinguishes itself by being N-methylated at this position, whereas C_2_ retains free amines.

Gentamicin has broad-spectrum activity, showing effectiveness against a wide variety of serious bacterial infections in several animal species that are caused by susceptible Gram-negative and some Gram-positive aerobic bacteria [3]. The mechanism of action of aminoglycosides involves several phases. These include binding of the drug to the components of bacterial cell membranes, resulting in displacement of divalent cations and increased membrane permeability. This allows for aminoglycoside entry and accumulation in the cell and to its primary intracellular target, the bacterial 30S ribosome, where it results in the mistranslation of proteins. This results in aminoglycoside accumulation, protein mistranslation, and synthesis inhibition, ultimately resulting in concentration-dependent bacterial killing [4,5,6].

Although gentamicin has been well-established in a variety of clinical settings, its use is also associated with adverse effects, as seen with other aminoglycosides, including nephrotoxicity and ototoxicity [4]. Human and animal studies have indicated that different components of gentamicin possess different potentials for toxicity, with some studies attributing the observed ototoxicity and nephrotoxicity of gentamicin mainly to the C_2_ component [7,8,9]. Various levels of the different components contained in different gentamicin formulations, as well as significant differences observed in some pharmacokinetic parameters between different gentamicin components, could have clinical or toxicological implications [9,10]. However, reported differences in the pharmacokinetics (PK) of gentamicin components are not consistent across species [2,10,11,12], emphasizing the importance of conducting PK studies of gentamicin components in species of interest such as piglets.

The PK of gentamicin has been studied in several species, including dogs [1], horses [2], chickens [10], fish [11], cats [13], pigs [14], sheep [15], calves [16], and buffalo calves [17]. Other than a single PK study reported several decades ago that used the intravenous (IV) route of administration [14], there are no gentamicin PK studies in piglets after the commonly used intramuscular (IM) or oral routes of administration.

Furthermore, most PK studies are conducted in healthy animals, and it is increasingly recognized that disease states—for example, bacterial infection and inflammation-mediated pathophysiological changes [18]—could affect the pharmacokinetic profile of several drugs to an extent that entails dosage adjustments.

Therefore, we designed the current study to address the following main objectives. Our first objective was to characterize the pharmacokinetics of the gentamicin components C_1_, C_1a_ and C_2_ in piglets after the commonly used IM route of administration and fill the knowledge gap in this area. Our second objective was to assess whether the health status of the piglets had any modifying effects on the disposition kinetics of gentamicin and determine the magnitude of any such effects by comparing the gentamicin PKs between healthy and infected piglets. To do so, and because bacterial infections in piglets occur more often as co-infections rather than as single infections [19], we developed and used a co-infection model induced by intranasal inoculation of *Actinobacillus pleuropneumoniae* (AP) and *Pasteurella multocida* (PM), which represent two important pathogens that are commonly associated with porcine respiratory disease complex (PRDC) [20]. Finally, it was our aim to perform limited pharmacokinetic/pharmacodynamic (PK/PD) analyses using gentamicin PK information obtained from the healthy and infected pigs and published minimum inhibitory concentration (MIC) distribution data to gain further understanding of our PK data in the context of the susceptibility patterns of the pathogens. This could assist in making informed decisions during antibiotic and dosage regimen selections.

## 2. Results

### 2.1. Confirmation of Infection

After 12 h post-infection, the piglets that were infected began to display prominent clinical signs such as lethargy, coughing, and minor respiratory challenges, distinguishing them from the healthy piglets. This confirmed the successful creation of a co-infected piglet model. The average body temperature for the healthy group stood at 37.23 ± 0.85 °C, while the infected group had a significantly higher temperature of 39.83 ± 0.38 °C (*p* < 0.01). In addition, PCR target gene amplification tests positively identified the apxIVA gene for AP and the kmt1 gene for PM in the infected piglets, with sizes of 377 bp (Figure 2A) and 460 bp (Figure 2B), respectively.

### 2.2. Determination of Plasma Gentamicin Concentrations Using Validated LC/MS

The LC/MS method was validated for several parameters. The linearity of the method was established with a standard curve, encompassing gentamicin concentrations from 0.1 µg/mL up to 50 µg/mL. The coefficients of linear regression (r^2^), slope and y-intercept for each gentamicin component are outlined in Table 1. For all the components, the R^2^ values were notably high, ranging from 0.998 to 1.000, suggesting excellent linearity. The slopes varied from 0.928 to 1.011, with C_1_, C_1a_ and C_2_ exhibiting intra-assay slopes of 0.928, 0.938 and 0.986 and inter-assay slopes of 0.996, 0.970 and 1.011, respectively. The intercepts ranged from −0.156 to 0.397. The limit of detection (LOD) values spanned from 0.008 to 0.111, whereas the limit of quantitation (LOQ) values were between 0.026 and 0.365 across the components and assay conditions.

The accuracy (recovery) and precision (relative standard deviation, RSD%) of the LC/MS method for the inter-assay and intra-assay runs for each component of gentamicin across the different specified concentrations are presented in Table 2. Nine nominal concentrations ranging from 0.1 μg/mL to 50 μg/mL were used in these assays.

The intra-assay accuracy study for gentamicin component C_1_ resulted in mean recovered concentrations with a range of 0.12 to 46.20 μg/mL, with an accuracy% ranging from 92% (for 50 μg/mL) to 115% (for 0.1 μg/mL), falling within an acceptable deviation of 8 to 15% from the tested nominal concentrations. The inter-assay accuracy results for C_1_ also showed acceptable deviations of 4 to 13%, with the mean recovered concentrations ranging between 0.11 μg/mL and 48.12 μg/mL. The precision of the C_1_ analysis was within an acceptable RSD of less than 11% for both the inter-assay and intra-assay runs.

For C_1a_, the intra-assay accuracy analysis showed deviations that fell within 10% of the nominal concentrations for eight out of nine (89%) of the concentration levels, with the exception of a 17% deviation observed for the 0.1 μg/mL concentration level. For the inter-assay analysis of C_1a_, the deviations from nominal concentrations were less than 7% for six out of nine (67%) of the concentration levels, less than 20% for two concentration levels and less than 25% for one concentration level. The precision of the C_1a_ analysis was also within an acceptable RSD of less than 15% for both the inter-assay and intra-assay runs.

For C_2_, the intra-assay accuracy analysis showed deviations that fell within 15% of the nominal concentrations for six out of nine (67%) of the concentration levels, less than 20% for two concentration levels and greater than 25% for one concentration level. For the inter-assay analysis of C_2_, the deviations from nominal concentrations were less than 15% for eight out of nine (89%) of the concentration levels and less than 20% for one concentration level. The precision of the C_2_ analysis was also within an acceptable RSD of less than 15% for both the inter-assay and intra-assay runs across all the concentration levels except for the intra-assay precision of the 0.2 μg/mL level, which resulted in an RSD of 16%.

Overall, the LC/MS method developed here was robust with an acceptable linearity, sensitivity, accuracy and precision. Because we used several concentration levels, calculation of unknown gentamicin concentrations in the plasma of the piglets was based on calibration curves with six or more concentration levels, with an acceptable accuracy within ±15% of the nominal concentrations and an acceptable precision of ±15% RSD. The concentration–time profiles of gentamicin, measured using LC/MS, are presented in Figure 3.

### 2.3. Pharmacokinetic Analysis

The pharmacokinetic (PK) parameters (mean ± SD) calculated for the gentamicin components (C_1_, C_1a_ and C_2_), after a single IM administration (10 mg/kg) in the healthy and infected piglets are provided in Figure 4 and Table 3. The gentamicin was rapidly absorbed in both the healthy and infected pigs, with a peak plasma concentration (C_max_) reached within 15–30 min in both groups. No differences were observed between the healthy and infected group in the time to reach C_max_ (T_max_), whereas the C_max_ value was greater (*p* < 0.05) in the healthy piglets compared to the infected animals for the C_1_ and C_2_ components. Similarly, the extent of gentamicin exposure in terms of the 24 h area under the plasma concentration–time curve (AUC_24h_) was greater (*p* < 0.05) in the healthy piglets compared to the infected animals for all the C_1_, C_1a_ and C_2_ components, with the highest significant difference (*p* < 0.001) observed for the C_1a_ component (115.19 ± 0.51 h∙µg/mL in the healthy versus 83.32 ± 8.62 h∙µg/mL in the infected group, Table 4). The gentamicin was eliminated faster in the infected animals compared to the healthy piglets, with greater (*p* < 0.05) terminal half-lives (T_1/2_) observed in the healthy animals compared to the infected animals for all the C_1_, C_1a_ and C_2_ components. The highest (more than 2-fold) difference was observed for the C_1a_ component (24.94 ± 7.32 h in the healthy group versus 11.94 ± 0.08 h in the infected group, Table 4). In agreement with the shorter T_1/2_ in the infected animals, the gentamicin mean residence time (MRT) was shorter (*p* < 0.05) and its apparent clearance (CL/F) was greater (*p* < 0.05) in the infected animals for C_1a_ and C_2_ components, whereas the differences for C_1_ did not reach significant levels. The apparent volume of distribution (Vz/F) of gentamicin was greater (*p* < 0.05) in the infected animals than in the healthy animals for all the C_1_, C_1a_ and C_2_ components.

We analyzed the contribution of each gentamicin component (C_1_, C_1a_, C_2_) for the total gentamicin exposure in terms of the total rate (C_max_) and total extent (AUC_24h_) of exposure in these animals. As shown in Table 4 below, the contributions of C_1_, C_1a_ and C_2_ for the total gentamicin exposure were generally comparable between the healthy and infected piglets and ranged roughly from 22 to 25%, 33 to 37% and 40 to 42%, respectively.

### 2.4. PK/PD Analysis

Pharmacokinetic/pharmacodynamic (PK/PD) analyses allow researchers to optimize the dosing regimens of antibacterial agents and preserve their utility. The three main PK/PD indices that have been established to predict the clinical efficacy of antibacterial drugs are the time during which the concentration of the drug falls over the minimum inhibitory concentration, MIC (T > MIC); the peak concentration to the MIC ratio (C_max_/MIC); and the ratio of the 24 h area under the concentration–time curve divided by the MIC (AUC_24h_/MIC) [21,22]. The PK/PD index and pharmacodynamic target associated with the efficacy of aminoglycosides, such as gentamicin, is the C_max_/MIC >10 [21]. We integrated the gentamicin PK information in our study, including the total C_max_ and total AUC_24h_, with recently published regional MIC distribution data for piglet *P. multocida* (PM) isolates [23,24]. As shown in Table 5, with the current dosing regimen, the PK/PD target for efficacy (C_max_/MIC > 10) could be exceeded for both the 50% and 90% MIC distributions (MIC_50_ and MIC_90_) of PM, with a greater magnitude in the healthy piglets (C_max_/MIC_90_, 23.4) compared to the infected (C_max_/MIC_90_, 20.1) piglets.

We could not perform similar analyses for other species, including AP, due to the scarcity of the current MIC distribution data from a representatively high number of isolates. We suggest that this PK information needs to be integrated with regional antibiotic susceptibility patterns of other bacterial isolates to make informed antibiotic and dosage regimen selections against piglet infections.

## 3. Discussion

Although gentamicin has long been used as a treatment for respiratory diseases, colibacillosis, peritonitis, urinary tract infection and sepsis in piglets [14,25], its intramuscular pharmacokinetics in piglets were largely unknown. Furthermore, while within human medicine, it is recognized that the PK of many compounds can be altered by the presence of inflammation or infection, far less is known about these relationships within the framework of veterinary medicine [26]. Herein, we conducted a comparative PK study of gentamicin after a single intramuscular injection in healthy piglets and piglets co-infected with two important respiratory pathogens that are commonly associated with porcine respiratory disease complex (PRDC).

After co-infection of healthy piglets with AP and PM, clear clinical symptoms were observed. The infecting bacteria were identified via PCR, confirming the successful establishment of the co-infection model. A total of 12 h after the infection, the appearance of significant clinical signs in the piglets indicated successful establishment of the infection model. However, the observed variations in how the individual piglets responded to the infection could be explained by differences in microbial loads [27], genetic predisposition [28], and environmental conditions [29]. The extent and timing of disease manifestation are directly affected by the microbial load, while an individual’s genetic background can determine their vulnerability or resilience to infection. Additionally, environmental influences, such as levels of stress and cleanliness, are key in shaping the immune response. These elements together lead to the varied responses seen in infectious diseases, underscoring the necessity of adopting a comprehensive approach in both research and the management of diseases.

The variability observed in the disease progression among the piglets infected with the *A. pleuropneumoniae* and *P. multocida* pathogens can be attributed to several critical factors, each contributing to the diverse outcomes seen in infectious disease models. One of the most significant factors is individual variations in the immune response [30]. Each piglet’s immune system reacts differently to infection, influenced by genetic predispositions, previous exposure, and their overall health status at the time of infection. These variations dictate the severity and speed of disease progression, with some animals able to mount more effective responses than others. Another pivotal factor is the differences in pathogen load [31]. The initial inoculum dose and the pathogen’s ability to replicate within the host vary significantly among individuals, directly impacting the disease’s progression rate. Factors such as the site of infection, the virulence of the pathogen strain, and the effectiveness of the host’s initial immune response can all influence the replication rate, leading to variability in the clinical outcomes among the infected piglets.

After intramuscular injection of gentamicin at a dose of 10 mg/kg in the healthy and co-infected piglets, the components of gentamicin (C_1_, C_1a_, C_2_) were analyzed via validated LC/MS. Linearity validation is a crucial step in analytical method validation, ensuring that an analytical procedure demonstrates proportionality between the test results and the concentration of the analyte within a given range [32]. A well-defined linear relationship confirms the method’s ability to produce results that are directly proportional to the concentration of the analyte in the sample. High R^2^ values are indicative of the method’s strong predictability and consistent performance over the specified range. The slope values further emphasize this linearity. Ideally, a perfect linear method would have a slope of 1 [33]. The obtained slopes, lying close to this ideal value, suggested that this method offered almost proportional response across the considered concentration range. The intercept values, which ideally should be close to zero when there is no inherent bias in the method at a zero concentration, show minor deviations, implying minimal systemic errors in the method. Furthermore, the LOD and LOQ provide insights into the method’s sensitivity [34]. Lower LOD values signify that even trace amounts of the analyte can be detected, while the LOQ values suggest that the lowest concentration level that can be quantitatively determined with an acceptable precision and accuracy. In Table 1, the presented results highlight the robust linearity of the methods for the analyzed components. The high R^2^ values, ranging from 0.998 to 1.000, for both the intra-assay and inter-assay settings strongly indicated a near-perfect linear relationship between the observed results and the expected analyte concentrations for components C_1_, C_1a_ and C_2_. The strong linearity, combined with the sensitivity metrics (LOD and LOQ), underscored the method’s reliability and suitability for its intended purpose, ensuring accurate and consistent results across the analytical range.

Table 2 elaborates the intra-assay and inter-assay variations of the LC/MS method for the three gentamicin components (C_1_, C_1a_, C_2_) at various nominal concentrations. For the gentamicin component C_1_, the intra-assay accuracy percentages remained fairly close to the nominal values, with variations only ranging from 92.41% to 116.68%. Such values suggest that the method’s reliability in a single experimental run is commendable. Similarly, the inter-assay accuracy for C_1_, which assesses the reproducibility across different experimental runs, showed values ranging from 96.24% to 113.64%. This tight range further confirms the method’s robustness and adaptability across different experimental conditions. For the gentamicin components C_1a_ and C_2_, a similar trend in accuracy was observed. The intra-assay accuracies for both components closely matched their nominal concentrations, suggesting good reliability within a single experimental session. Yet, it was noted that the precision for C_1a_ in the intra-assay had a wider range, especially compared to C_1_. This could indicate a slightly more variable consistency across repeated measurements for C_1a_ within the same run. The inter-assay results for C_1a_ and C_2_ further confirmed the method’s reproducibility, with accuracy percentages staying within a narrow range of the nominal values. The precision values for these components in the inter-assay were also within acceptable limits, highlighting consistent results across different experimental conditions. The results from both the intra-assay and inter-assay measurements reflect a high degree of accuracy for the method, indicating its reliability in reflecting the true values. While the precision showed some variability, especially within the same experimental run, the ranges observed were within the acceptable limits for many applications. This reinforces the confidence in using this method for measuring gentamicin components in various experimental settings.

Gentamicin PK in piglets was characterized by rapid absorption from the IM site and a long elimination half-life. We found that it was inappropriate to make extensive comparisons of the PK findings of this study with previous reports because of scarcity of PK studies in piglets and several additional layers that could confound our comparisons such as the different routes of administration (intravenous versus IM), bioanalytical methods (immunoassay versus LC/MS), dose, age of animals and species of animals used. For example, in a 1995 study, gentamicin was administered intravenously (IV) to newborn male piglets aged from 4 to 12 h at the time of administration and 42-day-old castrated male piglets [14]. In that study, the mean terminal half-life, mean residence time (MRT), volume of distribution and clearance were 5.19 h, 6.62 h, 785 mL/kg, and 121 L/h/kg in the newborn piglets and 3.50 h, 2.82 h, 474 mL/kg and 166 L/h/kg in the older group, respectively, indicating age-dependent gentamicin PK differences even after the same (IV) route of administration. The mean IM terminal half-life (17–24 h), MRT (6.7–8.0 h) and apparent volume of distribution (1756–2928 mL/kg) in the healthy piglets (Table 3) were all greater than the IV PK values reported in the study cited above, whereas the mean apparent clearance of 74–189 mL/kg in the healthy piglets in our study was comparable with the IV clearance in the above study. The terminal half-life of IM gentamicin in the piglets in our study was also much longer than those reported for other species, including the IV half-life of 1 h in beagle dogs [1], IV half-life of 2.4–4.3 h in horses [2] and IM half-life of 8.5 h in laying hens [12]. Such species-dependent and route-of-administration-dependent variability in drug clearance, as we suggested earlier, need to be identified by conducting PK studies of gentamicin in the species of interest so that tailored treatment strategies that ensure efficacy while minimizing side effects or the development of antibacterial drug resistance can be devised.

The observed differences in the gentamicin elimination rates between the healthy and diseased piglets could be attributed to the physiological changes that occur during infection [35]. Infections can significantly affect the body’s normal physiological processes, leading to alterations in blood flow, tissue perfusion, and permeability. These changes are pivotal in understanding the pharmacokinetics of medications in infected versus healthy states. For instance, an infection often triggers an inflammatory response, which can alter blood flow and the distribution of blood to various organs. This, in turn, affects how drugs are transported to sites of action or elimination. Enhanced blood flow to certain organs might accelerate drug clearance, while reduced perfusion in others could delay it. Moreover, infections can compromise the integrity of biological membranes, affecting tissue permeability [36]. This could lead to an increased or decreased distribution of drugs across tissues, influencing their bioavailability and elimination rates. Such physiological alterations could explain why gentamicin shows faster elimination in infected piglets. The drug’s distribution and clearance are likely affected by the systemic changes induced by the infection, leading to a shorter half-life in these animals. Understanding these changes is crucial for tailoring antibiotic therapies in veterinary medicine, ensuring effective dosing that accounts for the altered pharmacokinetic profile in diseased animals.

In contrast to a sheep study [15] where gentamicin’s gamma half-life extended significantly to 82.1 h after multiple IM doses, this study in piglets found that gentamicin was eliminated faster in the tinfected animals compared to the healthy ones. This was evidenced by significantly greater terminal half-lives (T_1/2_) in the healthy piglets across all the gentamicin components (C_1_, C_1a_ and C_2_), with the most pronounced difference observed in the C_1a_ component (24.94 ± 7.32 h in the healthy piglets versus 11.94 ± 0.08 h in the infected piglets, *p* < 0.05). This highlights how the infection status can markedly influence the gentamicin pharmacokinetics, in contrast to the sheep study’s focus on dosage regimen differences.

Differences were also observed in the PK parameters of the different gentamicin components (Figure 4, Table 3). While C_1a_ and C_2_ seemed to clear at a similar rate, the apparent clearance of C_1_ was different and was the highest. This held true for both the healthy and infected groups. Similarly, the apparent volumes of distribution of C_1a_ and C_2_ were comparable and were smaller than that of C_1_. These differences were more apparent in the healthy group but were overshadowed by the large within-group variability for C_1_ observed in the infected animals for both CL/F and Vz/F (Figure 3). No apparent differences were observed among the C_1_, C_1a_ and C_2_ components for the other PK parameters, including T_max_, T_1/2_ and MRT.

Our comparative PK analysis indicated that the rate and extent of gentamicin exposure were greater (greater C_max_ and AUC_24h_) in the healthy animals compared to the infected piglets. This was also reflected in the slower elimination (longer terminal half-life and smaller apparent clearance) observed in the healthy animals compared to the infected piglets. Combining the individual exposures of the three components (C_1_, C_1a_, C_2_), the total gentamicin C_max_ was 16.3% lower and the total gentamicin AUC_24h_ was 36.1% lower in the infected animals compared to the healthy piglets. These findings confirmed our initial hypothesis that the PK of gentamicin could be affected by the health status of the animals. Because our earlier comparative studies with tylosin, a macrolide antibiotic, also demonstrated significantly lower exposure (C_max_ and AUC_24h_) and shorter T_1/2_ values in co-infected pigs compared to healthy pigs [37], we assume this phenomenon to be common to several classes of antimicrobial agents. The clinical significance of these differences is, however, dependent on several factors. These may include the magnitude of the PK differences with respect to the susceptibility or MIC of the target pathogen, whether the antibacterial drug has time-dependent or concentration-dependent activity, and the type of PK parameter affected by the health status of the animals. For instance, in our study, gentamicin absorption after administration via the IM route was rapid, and T_max_ was not affected by the health status of the piglets. And, just like other aminoglycosides, maintaining an early gentamicin C_max_ of 10 times or higher than the MIC could be effective against susceptible pathogens like *P. multocida* with a MIC_90_ of 4 μg/mL, and this was achievable for both the healthy and infected piglets. Aminoglycosides exhibit concentration-dependent killing along with prolonged persistent effects that protect against regrowth when the active drug concentration falls below the MIC [21]. Therefore, the disease effects in later PK processes, such as a shorter elimination half-life in the infected piglets, would have minimal clinical implications. This is because the C_max_ was sufficiently higher than the target MIC, and the observed shorter half-life in the infected piglets was still long enough (greater than 11 h, Table 3) to support the once-per-day IM regimen of gentamicin.

A notable limitation of our study is the absence of a direct investigation into the antibiotic residues resulting from the use of gentamicin in the piglets. Despite the comprehensive analysis of gentamicin pharmacokinetics and its disposition in healthy versus infected piglets, we did not extend our research to examine the persistence of gentamicin residues in the animal tissues post-treatment. Recognizing the critical importance of such studies is essential, as they provide key insights into the potential for antibiotic residues to enter the food chain, posing risks to consumer health and contributing to the broader challenge of antimicrobial resistance.

## 4. Conclusions

This study aimed to investigate the intramuscular PKs of the gentamicin components C_1_, C_1a_ and C_2_ in piglets, evaluating how their health status affects the drug’s disposition. We found that gentamicin is rapidly absorbed from the IM injection site and has a lengthy elimination half-life. Notably, the PK profile of gentamicin was influenced by the piglets’ health, with the infected piglets showing reduced drug exposure and quicker elimination than the healthy ones. Additionally, the PK characteristics, such as their apparent clearance and volume of distribution, varied among the gentamicin components, with component C_1_ showing the highest values in both the healthy and infected piglets. The insights gained from this research, combined with knowledge of pathogen antibiotic resistance patterns, could inform the selection of appropriate antibiotic treatments and dosages, thereby extending the effectiveness of gentamicin in treating diseases in both humans and animals.

## 5. Materials and Methods

### 5.1. Chemicals and Reagents

The reference standards for gentamicin and nicotinamide adenine dinucleotide (NAD) were obtained from Sigma-Aldrich (St. Louis, MO, USA). The gentamicin injectable solution contained 50 mg of gentamicin per milliliter (mL). The solution also included 50% propylene glycol, and 4% benzyl alcohol was used as a preservative to inhibit bacterial growth. This injectable solution was obtained from Samyang Anipharm (Seoul, Republic of Korea). Additionally, for the purpose of cultivating bacteria, bacterial growth media, including brain heart infusion (BHI) and Mueller–Hinton broth (MHB) were purchased from BD Company (Franklin Lakes, NJ, USA). Furthermore, defibrinated sheep blood (SBD), essential for specific culture requirements, was purchased from Kisan Bio (Seoul, Republic of Korea).

### 5.2. Bacterial Culture and Confirmation of Infection

The bacterial strains, AP (BA2000022) and PM (BA1700099), used to induce bacterial infections were obtained from the Animal and Plant Quarantine Agency (Kimchen, Republic of Korea). Each displayed a minimum inhibitory concentration (MIC) in line with the MIC_90_ for gentamicin, both noting a MIC of 16 μg/mL. To achieve steady growth rates, these strains underwent three separate culture sessions. The AP was grown in BHI medium that was enriched with 0.02% NAD, while the PM strain was propagated in MHB with a 2% SBD additive. To keep track of the induced infections from these bacteria, nasal samples were taken using swabs procured from Copan Diagnostics (Murrieta, CA, USA). Once collected, these swabs were labeled, chilled using ice packs in coolers, and subsequently shipped to the research facility within a timeframe of 4 h. The identification and confirmation of the infections were facilitated using PCR techniques. Specifically, the apxIVA gene was targeted for AP [38], and the kmt1 gene was targeted for PM [39].

### 5.3. Animal Experimental Design

Twelve male piglets, a crossbreed of Duroc × Landrace × Yorkshire, around 5–6 weeks of age and weighing an average of 9.5 ± 1.1 kg, were acquired from Petobio (Hanam, Gyeonggi-do, Republic of Korea). They were then moved to the Gyeongsangbuk-do Veterinary Service Laboratory (Daegu, Republic of Korea). The research on these animals received approval from the Petobio Clinical Institute’s Animal Ethics Committee (PTB-2022-IACUC013-A). Upon reaching their new location, the piglets were allowed a week to adjust, with free access to food and water. After this period, they were assigned into two groups: six piglets were designated for infection and treatment (infected), and the other six were treated but not infected (healthy) (Figure 5).

The piglets were challenged as described previously [30]. A total of 40 mL of the bacterial blend was centrifuged at 3500 rpm for 10 min. The top liquid layer (supernatant) was then discarded. The leftover bacterial cells were combined with a 40 mL saline solution (0.9% NaCl). To administer the bacterial dose, the infected piglets were firmly restrained using a snare, making them sit in a posture similar to a seated dog, with their front limbs extended. As they inhaled, each piglet in this group received an intranasal application of the bacterial mix via a syringe. A total of 1 mL was dispensed and divided between both nostrils (0.5 mL for each nostril), consisting of 2.0 × 10^9^ CFU/mL each of AP (BA2000022) and PM (BA1700099).

### 5.4. Drug Administration and Sample Collection

12 h after the infection, both the healthy and infected groups received an intramuscular injection of gentamicin at a dose of 10 mg/kg, as described in the manufacturer’s instructions [40]. Blood samples were drawn from the jugular vein of each piglet at set intervals: before administration and then at 0.25, 0.5, 0.75, 1, 2, 4, 8, 12, 36 and 48 h post-injection. The samples were placed into anticoagulant-containing vacutainers supplied by BD Company (NJ, USA). Once the samples were centrifuged at 3000 rpm for 10 min, the plasma was separated, decanted into Eppendorf containers, and stored at −70 °C for future analysis.

### 5.5. Liquid Chromatography/Mass Spectrometry Analysis

The concentrations of gentamicin in the plasma from both the infected and non-infected animals were measured using a validated LC/MS approach [41]. To initiate the sample preparation, we mixed 90 µL of the plasma samples with 10 µL of a standard solution to ensure a consistent reference for all the samples. Following this, 100 µL of 0.5% formic acid in purified water was added to the mixture. This step was essential for precipitating proteins and facilitating the release of gentamicin from the plasma matrix. The solution was then vigorously mixed using a vortex mixer to ensure thorough integration of the components. Subsequently, 400 µL of acetonitrile, a solvent known for its efficiency in extracting a wide range of drugs including aminoglycosides such as gentamicin, was added to the mixture. This was again followed by vortex mixing to promote maximal extraction of the drug from the plasma. To further optimize the extraction process, 200 µL of pure water was added to adjust the solvent polarity, and the mixture was vortexed at a speed of 2500 rpm for 15 min. After achieving a homogenous solution, the sample underwent centrifugation at a speed of 12,000 rpm for 15 min. This high-speed centrifugation was critical for achieving a clear separation, allowing us to collect a pure supernatant devoid of plasma proteins and other potential contaminants. Finally, the top 70 µL of the clear supernatant was carefully extracted and subjected to LC/MS analysis for precise quantification of the gentamicin components.

The analysis was conducted using an Agilent Technologies HPLC 1200 series system (Santa Clara, CA, USA). The chromatographic process was performed using an Eclipse Plus C18 column (50 mm × 2.1 mm with a 1.8 μm particle size) from Agilent Technologies, and it was maintained at 40 °C. For the mobile phase, solution A comprised 2mM ammonium acetate in 0.1% formic acid with 5% acetonitrile, while solution B had the same acetate and formic acid contents but with 95% acetonitrile. The best gradient elution is shown in Table 6. A steady flow rate of 0.35 mL/min was maintained, and the injection volume was 5 µL.

The HPLC system was linked to an Agilent Technologies 6140 single Mass Spectrometer equipped with a multimode source. For this specific task, the ESI was set to the positive mode. It operated at a gas temperature of 250 °C and a vaporizer setting of 300 °C. Nitrogen served as the drying gas, with a flow setting of 5 L/min. Nitrogen was also used as the nebulizing gas at a pressure of 30 psi. The system had a capillary voltage set to 5000 V. The multiple reaction monitoring (MRM) had a dwell time of 200 ms, while the fragmentation voltage was set to 70 V. Additionally, a collision energy of 15 V was utilized.

Validation of the assay was conducted following a procedure outlined in a previous study [42]. For daily operations, new working standards were always established. Calibration curves were made using a range of solutions with concentrations between 0.1 µg/mL and 50 µg/mL. These curves were then used as references to gauge the amounts of analytes in the test samples. The proposed method’s efficacy was tested using plasma samples spiked at nine distinct concentrations. The limit of detection (LOD) was identified when the signal-to-noise ratio exceeded 3. The limit of quantitation (LOQ) was the point where this ratio surpassed 10 in the blank samples that had the analytes added.

### 5.6. Pharmacokinetic Study

The concentration of gentamicin in the plasma from each piglet over time was analyzed using Phoenix WinNonlin Version 8.3 software (Certara, Princeton, NJ, USA) using a non-compartmental approach. The peak drug concentration (C_max_) was directly read from the data, and the time to reach this peak (T_max_) was the moment when C_max_ was first noted. The area under the curve (AUC) was determined using the linear trapezoidal technique. Additional PK parameters, including the terminal half-life (T_1/2_) and the mean residence time (MRT), were also generated.

### 5.7. Statistical Analysis

The data are presented as mean values accompanied by standard deviations. The statistical analysis was conducted using Student’s *t*-tests using GraphPad Prism software version 8.0.1 (San Diego, CA, USA). A *p*-value below 0.05 was considered to be statistically significant.

## Figures and Tables

**Figure 1 antibiotics-13-00372-f001:**
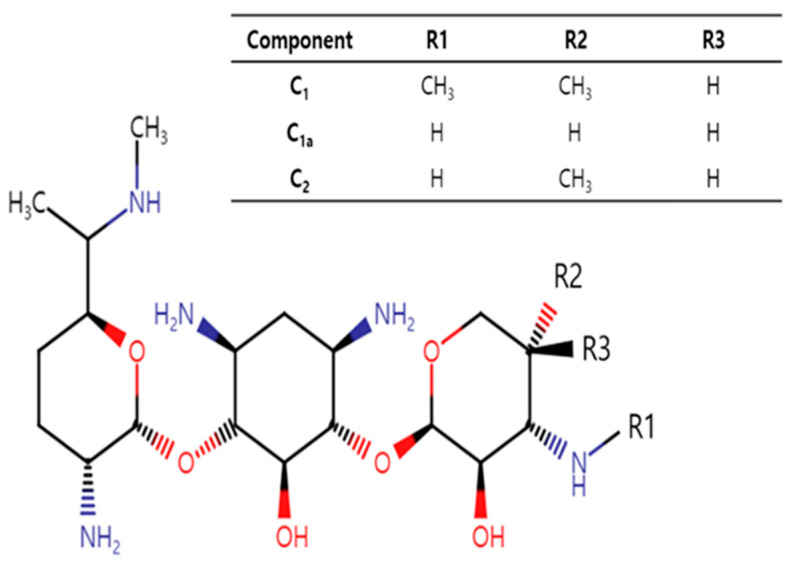
Structure of gentamicin C_1_, C_1a_ and C_2_, emphasizing their distinctive methylation patterns on the 2-amino-hexose ring.

**Figure 2 antibiotics-13-00372-f002:**
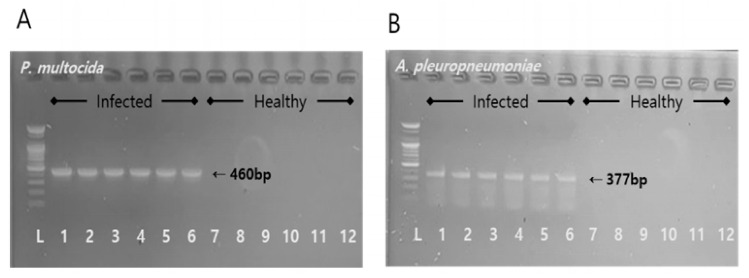
PCR target gene amplification results displaying the successful identification of the kmt1 gene for *Pasteurella multocida* at 460 bp (**A**) and apxIVA gene for *Actinobacillus pleuropneumoniae* at 377 bp (**B**) in the infected piglets.

**Figure 3 antibiotics-13-00372-f003:**
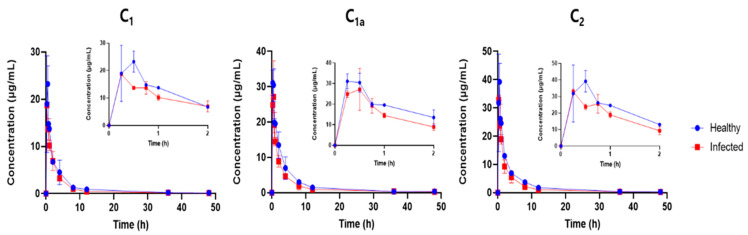
Concentration–time profiles (mean ± SD) of gentamicin components C_1_, C_1a_ and C_2_ after a single intramuscular administration in healthy (*n* = 6) and infected (*n* = 6) piglets. Inset graphs represent the first two hours when the differences between the healthy and infected groups were more apparent.

**Figure 4 antibiotics-13-00372-f004:**
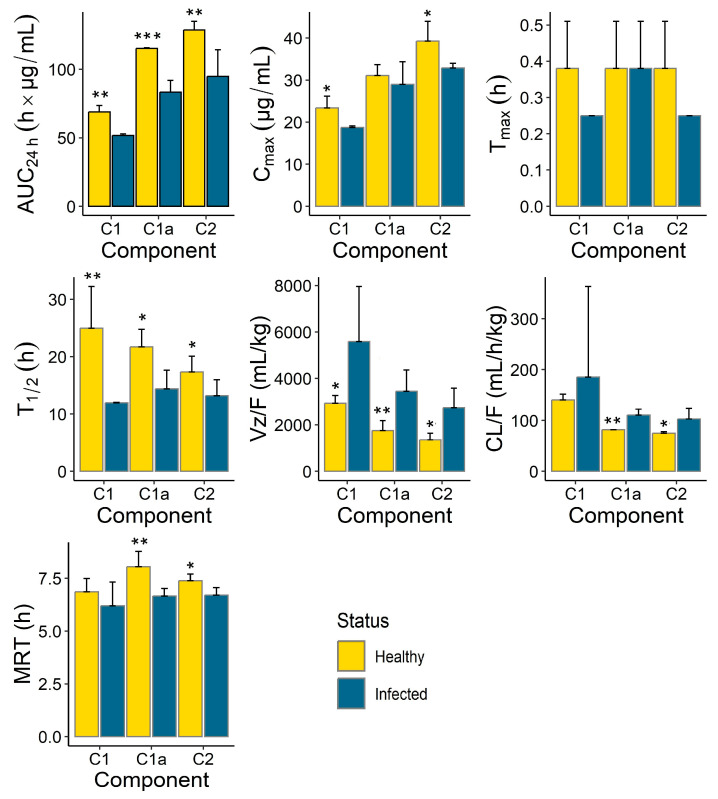
Pharmacokinetic parameters (mean ± SD) of gentamicin components C_1_, C_1a_ and C_2_ in healthy (*n* = 6) and infected (*n* = 6) piglets. Within each gentamicin component (C_1_, C_1a_, C_2_) and parameter, the differences between the healthy and infected animals are indicated as: *** *p* < 0.001, ** *p* < 0.01, * *p* < 0.05. Parameter names are as described in Table 3 below.

**Figure 5 antibiotics-13-00372-f005:**
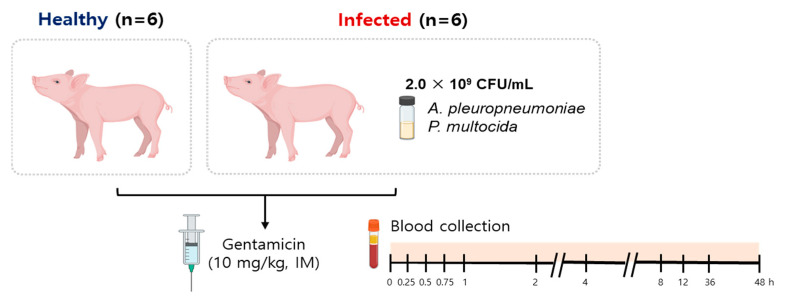
Animal experimental model. Six piglets were assigned for infection (infected), while the remaining six were treated without being exposed to infection (healthy). Twelve hours after the infection, both the healthy and infected groups were administered an intramuscular injection of gentamicin at a dose of 10 mg/kg. Blood samples were collected from the jugular vein of each piglet at specific intervals: prior to the administration and subsequently at 0.25, 0.5, 0.75, 1, 2, 4, 8, 12, 36 and 48 h post-injection.

**Table 1 antibiotics-13-00372-t001:** Linearity of the LC/MS method.

	Intra-Assay			Inter-Assay		
	C_1_	C_1a_	C_2_	C_1_	C_1a_	C_2_
Slope	0.928	0.938	0.986	0.996	0.970	1.011
Intercept	0.397	0.297	−0.156	0.229	0.267	0.098
R^2^	0.999	0.999	0.999	1.000	0.998	1.000
LOD	0.023	0.016	0.095	0.008	0.081	0.111
LOQ	0.076	0.054	0.313	0.026	0.267	0.365

LC/MS, liquid chromatography/mass spectrometry; LOD, limit of detection; LOQ, limit of quantitation.

**Table 2 antibiotics-13-00372-t002:** Intra-assay and inter-assay variations in LC/MS.

Gentamicin Component			Nominal Concentration (μg/mL)								
			0.1	0.2	0.5	1	2	5	10	20	50
C_1_	Intra-assay	Mean concentration (*n* = 6)	0.12	0.22	0.58	1.09	2.06	5.54	10.69	19.80	46.20
		Precision (RSD, %)	1.00	3.15	3.26	4.89	7.48	8.06	9.57	4.54	10.91
		Accuracy (%)	115.35	110.21	116.68	108.73	103.00	110.75	106.88	98.99	92.41
	Inter-assay	Mean concentration (*n* = 6)	0.11	0.23	0.56	1.04	1.96	5.42	9.99	20.50	48.12
		Precision (RSD, %)	10.50	7.71	8.13	2.42	1.73	10.73	1.61	0.57	4.77
		Accuracy (%)	111.52	113.64	112.53	103.59	98.03	108.49	99.94	102.50	96.24
C_1a_	Intra-assay	Mean concentration (*n* = 6)	0.12	0.22	0.54	1.09	2.00	5.14	10.94	19.39	46.81
		Precision (RSD, %)	4.82	2.75	13.03	5.66	13.05	0.43	6.08	7.58	3.89
		Accuracy (%)	117.40	109.06	107.23	109.18	99.89	102.82	109.45	96.96	93.61
	Inter-assay	Mean concentration (*n* = 6)	0.12	0.25	0.52	1.17	1.92	5.20	10.14	21.40	48.05
		Precision (RSD, %)	13.79	13.05	3.36	18.67	3.28	1.05	0.96	5.51	2.66
		Accuracy (%)	118.90	123.28	103.84	117.31	96.17	104.05	101.38	107.00	96.11
C_2_	Intra-assay	Mean concentration (*n* = 6)	0.12	0.23	0.49	1.27	1.72	4.83	9.35	18.19	49.73
		Precision (RSD, %)	10.85	16.32	1.89	10.32	5.70	2.22	4.93	8.95	4.36
		Accuracy (%)	118.07	116.24	98.01	127.18	86.07	96.59	93.48	90.95	99.47
	Inter-assay	Mean concentration (*n* = 6)	0.11	0.21	0.48	1.19	2.29	5.01	10.37	20.50	50.56
		Precision (RSD, %)	13.87	2.07	3.24	4.75	11.47	0.92	4.02	1.99	0.73
		Accuracy (%)	114.40	103.82	95.83	118.85	114.46	100.12	103.71	102.48	101.11

RSD, relative standard deviation.

**Table 3 antibiotics-13-00372-t003:** PK parameters (mean ± SD) of gentamicin after a single IM dose (10 mg/kg) in healthy and infected piglets ^a^.

Parameter	C_1_		C_1a_		C_2_	
	Healthy	Infected	Healthy	Infected	Healthy	Infected
T_1/2_ (h)	24.94 ± 7.32 **	11.94 ± 0.08	21.72 ± 3.06 *	14.38 ± 3.26	17.34 ± 2.75 *	13.18 ± 2.82
T_max_ (h)	0.38 ± 0.13	0.25 ± 0.00	0.38 ± 0.13	0.38 ± 0.13	0.38 ± 0.13	0.25 ± 0.00
C_max_ (μg/mL)	23.35 ± 2.83 *	18.70 ± 0.41	31.10 ± 2.59	29.01 ± 5.33	39.30 ± 4.67 *	32.88 ± 1.13
AUC_24h_ (h∙μg/mL)	68.77 ± 4.82 **	51.65 ± 1.19	115.19 ± 0.51 ***	83.32 ± 8.62	128.72 ± 6.25 **	94.82 ± 19.52
Vz/F (mL/kg)	2928.66 ± 338.05 *	5589.60 ± 2374.77	1756.42 ± 430.35 **	3449.78 ± 919.74	1357.81 ± 280.39 *	2739.07 ± 845.51
CL/F (mL/h/kg)	139.97 ± 11.43	185.38 ± 178.25	81.94 ± 0.67 **	110.80 ± 11.39	74.93 ± 2.86 *	102.85 ± 20.84
MRT (h)	6.86 ± 0.63	6.19 ± 1.13	8.04 ± 0.72 **	6.66 ± 0.36	7.39 ± 0.30 *	6.70 ± 0.36

T_1/2,_ half-life; Tmax, the time to reach the maximum plasma concentration; C_max_, maximum plasma concentration; AUC_24h_, area under the plasma concentration–time curve within 24 h; Vz/F, apparent volume of distribution; CL/F, apparent clearance; MRT, mean residence time; PK, pharmacokinetic; IM, intramuscular. Within each gentamicin component (C_1_, C_1a_, C_2_) and parameter, the differences between the healthy and infected animals are indicated as: *** *p* < 0.001, ** *p* < 0.01, * *p* < 0.05. ^a^ *n* = 6 per group.

**Table 4 antibiotics-13-00372-t004:** Contribution of C_1_, C_1a_ and C_2_ for total gentamicin exposure.

Status	Component	% of Total C_max_	% of Total AUC_24h_
Healthy	C_1_	24.9	22.0
	C_1a_	33.2	36.8
	C_2_	41.9	41.2
	Total	100.0	100.0
Infected	C_1_	23.2	22.5
	C_1a_	36.0	36.3
	C_2_	40.8	41.3
	Total	100.0	100.0

**Table 5 antibiotics-13-00372-t005:** PK/PD predictors of gentamicin activity against *P. multocida*.

Status	Total C_max_ (μg/mL)	Total AUC_24h_ (h·μg/mL)	MIC (μg/mL)	C_max_/MIC	AUC_24h_/MIC	MIC Source ^a^
Healthy	93.8	312.7	MIC_50_ = 2	46.9	156.3	[23,24]
			MIC_90_ = 4	23.4	78.1	
Infected	80.6	229.8	MIC_50_ = 2	40.3	114.9	
			MIC_90_ = 4	20.1	57.4	

PK, pharmacokinetics; PD, pharmacodynamics; MIC, minimum inhibitory concentration; MIC_50_, the MIC for 50% of the tested bacterial isolates; MIC_90_, the MIC for 90% of the tested bacterial isolates; ^a^ MIC sources: one study with a total of 454 isolates of *P multocida* collected from all the provinces of Korea between 2010 and 2016 and another study with a total of 240 *P. multocida* isolated from pneumonic pigs in Korea between 2008 and 2016.

**Table 6 antibiotics-13-00372-t006:** Gradient condition of the mobile phase.

Time (min)	A ^a^ (%)	B ^b^ (%)
0	80	20
1	5	95
1.3	5	95
4	80	20
4.1	80	20
5	80	20

^a^ 2mM ammonium acetate in 0.1% formic acid, 5% acetonitrile. ^b^ 2mM ammonium acetate in 0.1% formic acid, 95% acetonitrile.

## Data Availability

The data are available from the corresponding author upon reasonable request.

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
