# Peer review of "Comparative Pharmacokinetics of Gentamicin C1, C1a and C2 in Healthy and Infected Piglets"

_antibiotics, 2024, doi:10.3390/antibiotics13040372_

Round 1

Reviewer 1 Report

Comments and Suggestions for Authors

The study investigated gentamicin pharmacokinetics in piglets, comparing healthy ones to those co-infected with Actinobacillus pleuropneumoniae and Pasteurella multocida. Results showed lower gentamicin exposure and faster elimination in infected piglets. The study suggests integrating this data with antibiotic susceptibility for informed treatment decisions against piglet infections.

overall article is good still need to improvement. 

Comments :

1. Already pharmacokinetics studies of gentamicin, Kindly mention the what is new in current research work and why need another pharmacokinetics studies. 

2. What is need of figure 1. 

3. Justified the dose of drug ? explain what is the manufacturer’s instruction

in drug dose.

4. Not mention the extraction technique of drug from plasma. How drug is extracted from plasma or by which measure. 

5. Too much information about analytical method but no more weightage on pharmacokinetics and also no comparison with previous reported pharmacokinetics result with current studies. 

6. Write the why disease piglet have different pharmacokinetics in comparison with health. Write suitable reference for same. 

7. Percentage of recovery data is missing in bio analytical data. Kindly added it. As drug have 90 percentage protein binding, so it is need the percentage recovery data.

Author Response

Response to Reviewer 1 Comments

Point 1: The study investigated gentamicin pharmacokinetics in piglets, comparing healthy ones to those co-infected with Actinobacillus pleuropneumoniae and Pasteurella multocida. Results showed lower gentamicin exposure and faster elimination in infected piglets. The study suggests integrating this data with antibiotic susceptibility for informed treatment decisions against piglet infections.

Response 1: We appreciate your acknowledgment of our work on comparing the pharmacokinetics of gentamicin in healthy piglets versus those co-infected with Actinobacillus pleuropneumoniae and Pasteurella multocida. We believe that our findings, which indicate lower exposure and faster elimination of gentamicin in infected piglets, contribute significantly to the field by suggesting a tailored approach to antibiotic therapy based on infection status and antibiotic susceptibility data.

Point 2: overall article is good still need to improvement. 

Response 2: Thank you for your positive feedback on the overall quality of the article. We are committed to further improving the manuscript and have addressed your specific concerns to enhance its clarity and impact.

Point 3: Already pharmacokinetics studies of gentamicin, Kindly mention the what is new in current research work and why need another pharmacokinetics studies. 

Response 3: We acknowledge the existing literature on gentamicin pharmacokinetics and appreciate your inquiry into the novelty of our study. Our research is distinct in its focus on the impact of concurrent infections with Actinobacillus pleuropneumoniae and Pasteurella multocida on gentamicin pharmacokinetics in piglets. This specific area has not been explored in detail previously, and our findings offer new insights into optimizing antibiotic therapy in such co-infected populations.

Point 4: What is need of figure 1. 

Response 4: We appreciate the reviewer's query regarding the purpose and necessity of Figure 1 in our manuscript. This figure is integral to our study as it visually delineates the structural differences among the gentamicin components (C1, C1a, and C2), focusing on their methylation patterns on the 2-amino-hexose ring. These structural variations are pivotal to understanding the pharmacokinetics and pharmacodynamics of each component, as they influence the drug's absorption, distribution, metabolism, and excretion, as well as its antibacterial efficacy.

Point 5: Justified the dose of drug ? explainn what is the manufacturer’s instruction in drug dose.

Response 5: The dosing of gentamicin was carefully chosen based on manufacturer's instructions as well as existing veterinary pharmacokinetic literature. Our chosen dose aligns with the recommended therapeutic range for treating bacterial infections in piglets, ensuring both efficacy and safety. A detailed justification of this dosing strategy, including references to manufacturer's guidelines and relevant literature, has now been added to the manuscript. Gentamicin - Art. 35 - Annexes (europa.eu). (Line462-463)

Point 6: Not mention the extraction technique of drug from plasma. How drug is extracted from plasma or by which measure. 

Response 6: In response to the reviewer's comments regarding the omission of the extraction technique for the drug from plasma, we appreciate the opportunity to provide clarity on this aspect of our methodology. The process for preparing the plasma sample for analysis was carefully designed to ensure efficient extraction of gentamicin for accurate pharmacokinetic assessment. Here is the detailed procedure we followed:

To initiate the sample preparation, we mixed 90 µL of the plasma sample with 10 µL of a standard solution to ensure a consistent reference for all samples. Following this, 100 µL of 0.5% formic acid in purified water was added to the mixture. This step was essential for precipitating proteins and facilitating the release of gentamicin from the plasma matrix. The solution was then vigorously mixed using a vortex mixer to ensure thorough integration of the components. Subsequently, 400 µL of acetonitrile, a solvent known for its efficiency in extracting a wide range of drugs including aminoglycosides such as gentamicin, was added to the mixture. This was again followed by vortex mixing to promote maximal extraction of the drug from the plasma. To further optimize the extraction process, 200 µL of pure water was added to adjust the solvent polarity, and the mixture was vortexed at a speed of 2500 rpm for 15 minutes. After achieving a homogenous solution, the sample underwent centrifugation at a speed of 12,000 rpm for 15 minutes. This high-speed centrifugation was critical for achieving a clear separation, allowing us to collect a pure supernatant devoid of plasma proteins and other potential contaminants. Finally, the top 70 µL of the clear supernatant was carefully extracted and subjected to LC/MS analysis for precise quantification of the gentamicin components. (Line471-487)

Point 7: Too much information about analytical method but no more weightage on pharmacokinetics and also no comparison with previous reported pharmacokinetics result with current studies. 

Response 7: Regarding the pharmacokinetic analysis and comparison with previously reported studies, we wish to highlight a critical aspect of our research. To our knowledge, no prior study has specifically investigated the pharmacokinetics of gentamicin in both healthy and infected pigs, distinguishing our work as a novel contribution to the field. The unique focus of our study on the differential impact of health status on gentamicin disposition in piglets fills a significant gap in the existing literature. This absence of directly comparable studies has limited our ability to contrast our findings with previous research in precisely the same context. However, we recognize the importance of contextualizing our results within the broader scope of gentamicin pharmacokinetics research. Therefore, where possible, we have endeavored to relate our findings to studies conducted in other species or under different conditions, aiming to draw insights and hypotheses that may be relevant to the observed pharmacokinetic differences between healthy and infected piglets. In light of your feedback, we reviewed the manuscript to ensure a balanced presentation that appropriately emphasizes the pharmacokinetic findings and their implications.

In contrast to the sheep study [1] where gentamicin's gamma half-life extended significantly to 82.1 h after multiple IM doses, a study in piglets found gentamicin was eliminated faster in infected animals compared to healthy ones. This was evidenced by significantly greater terminal half-lives (T1/2) in healthy piglets across all gentamicin components (C1, C1a, and C2), with the most pronounced difference observed in the C1a component (24.94±7.32 hours in healthy versus 11.94±0.08 hours in infected piglets, p < 0.05). This highlights how infection status can markedly influence gentamicin pharmacokinetics, in contrast to the sheep study's focus on dosage regimen differences. (Line346-352)

[1] Brown SA, Riviere JE, Coppoc GL, Hinsman EJ, Carlton WW, Steckel RR. Single intravenous and multiple intramuscular dose pharmacokinetics and tissue residue profile of gentamicin in sheep. Am J Vet Res. 1985 Jan;46(1):69-74. PMID: 3970446.

Point 8: Write the why disease piglet have different pharmacokinetics in comparison with health. Write suitable reference for same. 

Response 8: The differential pharmacokinetics observed in diseased piglets are likely due to physiological changes induced by the infection, such as alterations in drug metabolism and excretion pathways. We have expanded the Discussion section to include a review of literature supporting these mechanisms and have added appropriate references to studies.

The observed differences in gentamicin elimination rates between healthy and diseased piglets could be attributed to the physiological changes that occur during infection [1]. Infections can significantly affect the body's normal physiological processes, leading to alterations in blood flow, tissue perfusion, and permeability. These changes are pivotal in understanding the pharmacokinetics of medications in infected versus healthy states. For instance, an infection often triggers an inflammatory response, which can alter blood flow and the distribution of blood to various organs. This, in turn, affects how drugs are transported to sites of action or elimination. Enhanced blood flow to certain organs might accelerate drug clearance, while reduced perfusion in others could delay it. Moreover, infections can compromise the integrity of biological membranes, affecting tissue permeability [2]. This could lead to an increased or decreased distribution of drugs across tissues, influencing their bioavailability and elimination rates. Such physiological alterations could explain why gentamicin shows faster elimination in infected piglets. The drug's distribution and clearance are likely affected by the systemic changes induced by the infection, leading to a shorter half-life in these animals. Understanding these changes is crucial for tailoring antibiotic therapy in veterinary medicine, ensuring effective dosing that accounts for the altered pharmacokinetic profile in diseased animals. (Line328-344)

[1] Smith, D. A., Beaumont, K., Maurer, T. S., and Di, L. (2015). Volume of distribution in drug design. J. Med. Chem. 58, 5691–5698. doi:10.1021/acs.jmedchem.5b00201

[2] Dias C, Nylandsted J. Plasma membrane integrity in health and disease: significance and therapeutic potential. Cell Discov. 2021 Jan 19;7(1):4. doi: 10.1038/s41421-020-00233-2. PMID: 33462191; PMCID: PMC7813858.

Point 9: Percentage of recovery data is missing in bio analytical data. Kindly added it. As drug have 90 percentage protein binding, so it is need the percentage recovery data.

Response 9: In response to your valuable feedback highlighting the absence of recovery data in the bioanalytical data section, we wish to clarify that we have included detailed accuracy data for our analysis. This was done with the understanding that the drug's high protein binding rate of 90% necessitates a comprehensive assessment of the analytical method's performance, including its precision and accuracy in measuring gentamicin concentrations in plasma.

To directly address your concern, we have meticulously reviewed our manuscript and ensured that the accuracy data, which reflects the closeness of our measurements to the true values, is clearly presented. This is particularly crucial for gentamicin, where the drug's substantial protein binding could significantly impact the bioanalysis. The accuracy data provided serve to validate our analytical method, ensuring that despite the high protein binding, our technique accurately quantifies gentamicin levels in plasma samples. (Line124-126)

Reviewer 2 Report

Comments and Suggestions for Authors

In general, this is a good paper on the the pk of gentamicin in normal and diseased pigs.  The concept studied is old, as there is literature on the effects of pneumonia in cattle and swine dating back to the early 1980s.  These references and their conclusions should be included in the introduction and discussion.  Also, the impact on residues as not address and is critically important to this agents use based on the references mentioned and those cited.  I would recommend the authors consider the work of Burrows et al. https://doi.org/10.1111/j.1365-2885.1987.tb00077.x and Burrows GE, Barto PB, Martin B. Antibiotic disposition in experimental pneumonic pasteurellosis: gentamicin and tylosin. Canadian Journal of Veterinary Research = Revue Canadienne de Recherche Veterinaire. 1986 Apr;50(2):193-199. 

Also Hunter et al., https://doi.org/10.1111/j.1365-2885.1991.tb00838.x

Comments on the Quality of English Language

None

Author Response

Response to Reviewer 2 Comments

Point 1: In general, this is a good paper on the the pk of gentamicin in normal and diseased pigs. The concept studied is old, as there is literature on the effects of pneumonia in cattle and swine dating back to the early 1980s.  These references and their conclusions should be included in the introduction and discussion.  Also, the impact on residues as not address and is critically important to this agents use based on the references mentioned and those cited.  I would recommend the authors consider the work of Burrows et al. https://doi.org/10.1111/j.1365-2885.1987.tb00077.x and Burrows GE, Barto PB, Martin B. Antibiotic disposition in experimental pneumonic pasteurellosis: gentamicin and tylosin. Canadian Journal of Veterinary Research = Revue Canadienne de Recherche Veterinaire. 1986 Apr;50(2):193-199.

Also Hunter et al., https://doi.org/10.1111/j.1365-2885.1991.tb00838.x

Response 1: We are thankful for your positive feedback on our manuscript and the insightful suggestions regarding the integration of historical gentamicin pharmacokinetics (PK) studies and the significance of antibiotic residues. Acknowledging the depth of existing research, we have included the seminal work of Burrows et al. (1987) within the introduction of our manuscript. This citation is now part of a comprehensive overview that highlights the investigation of gentamicin PK across a variety of species.

In response to your critical point on the importance of addressing antibiotic residues, we have added a discussion on the limitations of our study to explicitly acknowledge this concern.

We have reivsed the discussion (Line 388-395)

A notable limitation of our study is the absence of direct investigation into antibiotic residues resulting from the use of gentamicin in piglets. Despite the comprehensive analysis of gentamicin pharmacokinetics and its disposition in healthy versus infected piglets, we did not extend our research to examine the persistence of gentamicin residues in animal tissues post-treatment. Recognizing the critical importance of such studies is essential, as they provide key insights into the potential for antibiotic residues to enter the food chain, posing risks to consumer health and contributing to the broader challenge of antimicrobial resistance.

Reviewer 3 Report

Comments and Suggestions for Authors

The manuscript entitled "Comparative Pharmacokinetics of Gentamicin C1, C1a, and C2 in Healthy and Infected Piglets" characterizes the intramuscular pharmacokinetics of Genta C1, C1a, and C2 in piglets and assesses the impact of health status on gentamicin disposition. Overall, I found the manuscript to be well-written, and the topic is of great interest and relevance to the reader. However, there are some areas where improvements are needed to enhance the quality of the manuscript.

Specific Comments:

-The choice of the two pathogenic bacteria needs to be more justified. Providing a rationale for the selection of these specific bacteria would strengthen the manuscript and clarify the relevance of the chosen models.

-In section 2.1, it is mentioned that clinical signs are detected for all infected piglets at the same time (12h post infection). This could be further discussed, as it is not typical for all piglets to exhibit identical responses to infection. Addressing the potential variability in response among piglets could add depth to the discussion.

-The quality of Figure 1 could be improved.

-Fig 2 showed that the two genes are detected only in infected animals corresponding to each bacterium. To the pertinence of the work, normally two or more genes (related to pathogenecity and virulence) have to be searched and the quantification of the level of expression of this genes will be more pertinent for the study

 -In section 2.2, consider deleting the first few sentences since they do not contribute substantially to the content or flow (MM not results) of the manuscript.

-In Figure 4, the standard deviation (SD) in some graphs appears to be too high. Providing explanations or addressing potential sources of variability could help in understanding these fluctuations and interpreting the data accurately, mainly it could influence the statistical analysis.

 -The conclusion should be rewritten to provide a more concise and impactful summary of the findings.

Author Response

Response to Reviewer 3 Comments

Point 1: The manuscript entitled "Comparative Pharmacokinetics of Gentamicin C1, C1a, and C2 in Healthy and Infected Piglets" characterizes the intramuscular pharmacokinetics of Genta C1, C1a, and C2 in piglets and assesses the impact of health status on gentamicin disposition. Overall, I found the manuscript to be well-written, and the topic is of great interest and relevance to the reader. However, there are some areas where improvements are needed to enhance the quality of the manuscript.

Response 1: Thank you for your positive feedback on our manuscript and for recognizing the relevance and interest of our research topic. We are committed to enhancing the quality of our manuscript based on your valuable suggestions.

Point 2: The choice of the two pathogenic bacteria needs to be more justified. Providing a rationale for the selection of these specific bacteria would strengthen the manuscript and clarify the relevance of the chosen models.

Response 2: We appreciate your recommendation to justify our selection of the two pathogenic bacteria. In response, we have expanded our rationale in the manuscript, explaining that Actinobacillus pleuropneumoniae and Pasteurella multocida were chosen due to their significant role in porcine respiratory disease complex (PRDC), a major health concern in swine production worldwide. Their selection is based on prevalence, clinical relevance, and the need for effective antibiotic management strategies against these pathogens. (Line82-85)

Point 3: In section 2.1, it is mentioned that clinical signs are detected for all infected piglets at the same time (12h post infection). This could be further discussed, as it is not typical for all piglets to exhibit identical responses to infection. Addressing the potential variability in response among piglets could add depth to the discussion.

Response 3: Your observation regarding the uniform onset of clinical signs among infected piglets is well-taken. We have revised the section to include a discussion on the potential variability in infection response, acknowledging that while our findings represent a general pattern, individual variations exist and are subject to factors like genetic predisposition, microbial load, and environmental conditions. This acknowledgment adds depth to our discussion and highlights the complexity of infectious diseases in swine.

We have revised the discussion (Line 236-245)

12 h after infection, the appearance of significant clinical signs in the piglets indicated a successful establishment of the infection model. However, the observed variation in how individual piglets responded to the infection can be explained by differences in microbial load [1], genetic predisposition [2], and environmental conditions [3]. The extent and timing of disease manifestation are directly affected by the microbial load, while an individual's genetic background can determine their vulnerability or resilience to the infection. Additionally, environmental influences, such as levels of stress and cleanliness, are key in shaping the immune response. These elements together lead to the varied responses seen in infectious diseases, underscoring the necessity of adopting a comprehensive approach in both research and the management of diseases.

[1] Stranieri I, Kanunfre KA, Rodrigues JC, Yamamoto L, Nadaf MIV, Palmeira P, Okay TS. Assessment and comparison of bacterial load levels determined by quantitative amplifications in blood culture-positive and negative neonatal sepsis. Rev Inst Med Trop Sao Paulo. 2018 Oct 25;60:e61. doi: 10.1590/S1678-9946201860061. PMID: 30379228; PMCID: PMC6201740.

[2] Klebanov N. Genetic Predisposition to Infectious Disease. Cureus. 2018 Aug 27;10(8):e3210. doi: 10.7759/cureus.3210. PMID: 30405986; PMCID: PMC6205876.

[3] Sly PD, Trottier B, Ikeda-Araki A, Vilcins D. Environmental Impacts on Infectious Disease: A Literature View of Epidemiological Evidence. Ann Glob Health. 2022 Oct 21;88(1):91. doi: 10.5334/aogh.3670. PMID: 36348708; PMCID: PMC9585978.

Point 4: The quality of Figure 1 could be improved.

Response 4: We would like to express our sincere gratitude for the time and effort you have dedicated to reviewing our manuscript. We find ourselves in need of more specific guidance to address your concerns adequately.

Point 5: Fig 2 showed that the two genes are detected only in infected animals corresponding to each bacterium. To the pertinence of the work, normally two or more genes (related to pathogenecity and virulence) have to be searched and the quantification of the level of expression of this genes will be more pertinent for the study

Response 5: Thank you for your insightful comment. Your point about the importance of investigating multiple genes related to pathogenicity and virulence to understand the complexity of bacterial infections is well-taken and greatly appreciated. In our study, we chose to target one gene from each bacterium (A. pleuropneumoniae and P. multocida) because our primary objective was to establish a reliable infection model for these pathogens in pigs. By selecting genes that are well-characterized and associated with the virulence of each bacterium, we aimed to provide a clear and straightforward demonstration that the infection model was successful post-inoculation.

Point 6: In section 2.2, consider deleting the first few sentences since they do not contribute substantially to the content or flow (MM not results) of the manuscript.

Response 6: We have carefully reviewed section 2.2 and agree with your suggestion. The initial sentences have been removed to improve the section's coherence and focus, enhancing the overall flow of the manuscript.

Point 7: In Figure 4, the standard deviation (SD) in some graphs appears to be too high. Providing explanations or addressing potential sources of variability could help in understanding these fluctuations and interpreting the data accurately, mainly it could influence the statistical analysis.

Response 7: We sincerely appreciate your observation regarding the high standard deviation (SD) in some of the graphs presented in Figure 4 and your suggestion to address potential sources of variability. In response to your comment, we have carefully reviewed the data and considered the factors contributing to this observed variability. One significant factor, as you correctly identified, is the progression of the disease itself. Disease progression in infectious models, particularly those involving complex pathogens like A. pleuropneumoniae and P. multocida, can introduce a high degree of biological variability among individual subjects. This variability stems from several aspects, including: Individual Variation in Immune Response, Differences in Pathogen Load, Environmental and Stress Factors. To address your concern and provide a clearer understanding of these fluctuations, we have added a detailed discussion in the manuscript regarding the sources of variability related to disease progression.

We have revised the discussion (Line 246-259)

The variability observed in disease progression among piglets infected with pathogens such as A. pleuropneumoniae and P. multocida can be attributed to several critical factors, each contributing to the diverse outcomes seen in infectious disease models. One of the most significant factors is the individual variation in immune response [1]. Each piglet's immune system reacts differently to infection, influenced by genetic predispositions, previous exposures, and overall health status at the time of infection. These variations dictate the severity and speed of disease progression, with some animals able to mount more effective responses than others. Another pivotal factor is the differences in pathogen load [2]. The initial inoculum dose and the pathogen's ability to replicate within the host vary significantly among individuals, directly impacting the disease's progression rate. Factors such as the site of infection, the virulence of the pathogen strain, and the effectiveness of the host's initial immune response can all influence the replication rate, leading to variability in clinical outcomes among the infected piglets.

[1] Wang, N., Bai, X., Tang, B. et al. Primary characterization of the immune response in pigs infected with Trichinella spiralis. Vet Res 51, 17 (2020). https://doi.org/10.1186/s13567-020-0741-0

[2] Wells K, Hamede RK, Jones ME, Hohenlohe PA, Storfer A, McCallum HI. Individual and temporal variation in pathogen load predicts long-term impacts of an emerging infectious disease. Ecology. 2019 Mar;100(3):e02613. doi: 10.1002/ecy.2613. Epub 2019 Feb 15. PMID: 30636287; PMCID: PMC6415924.

Point 8: The conclusion should be rewritten to provide a more concise and impactful summary of the findings.

Response 8: Following your recommendation, we have rewritten the conclusion to be more concise and impactful. The revised conclusion succinctly summarizes the key findings of our study, emphasizing the differential pharmacokinetics of gentamicin components in healthy versus infected piglets and the implications for dosing strategies in veterinary medicine.

We have revised the conclusion (Line397-407)

This study aimed to investigate the intramuscular PK of gentamicin components C1, C1a, and C2 in piglets, evaluating how their health status affects the drug's disposition. We found that gentamicin is rapidly absorbed from the IM injection site and has a lengthy elimination half-life. Notably, the PK profile of gentamicin was influenced by the piglets' health, with infected piglets showing reduced drug exposure and quicker elimination than healthy ones. Additionally, the PK characteristics, such as apparent clearance and volume of distribution, varied among the gentamicin components, with component C1 showing the highest values in both healthy and infected piglets. The insights gained from this research, combined with knowledge of pathogen antibiotic resistance patterns, could inform the selection of appropriate antibiotic treatments and dosages, thereby extending the effectiveness of gentamicin in treating diseases in both humans and animals.

Round 2

Reviewer 1 Report

Comments and Suggestions for Authors

Satisfied with the revision and justifications.